# Relationship between no-visitation policy and the development of delirium in patients admitted to the intensive care unit

**Fumihide Shinohara**[1,2], **Takeshi Unoki**[1]*, **Megumi Horikawa**[2]

**1** Department of Acute and Critical Care Nursing, School of Nursing, Sapporo City University, Sapporo, Hokkaido, Japan, **2** Intensive Care Unit, Kin-ikyo Chuo Hospital, Sapporo, Hokkaido, Japan

* iwhyh1029@gmail.com

## Abstract

### Background

Due to the severe acute respiratory syndrome coronavirus 2 (SARS-Cov-2) pandemic, many hospitals imposed a no-visitation policy for visiting patients in hospitals to prevent the transmission of SARS-CoV-2 among visitors and patients. The objective of this study was to investigate the association between the no-visitation policy and delirium in intensive care unit (ICU) patients.

### Methods

This was a single-center, before-after comparative study. Patients were admitted to a mixed medical-surgical ICU from September 6, 2019 to October 18, 2020. Because no-visitation policy was implemented on February 26, 2020, we compared patients admitted after this date (after phase) with the patients admitted before the no-visitation policy (before phase) was implemented. The primary outcome was the incidence of delirium during the ICU stay. Cox regression was used for the primary analysis and was calculated using hazard ratios (HRs) and 95% confidence intervals (CIs). Covariates were age, sex, APACHE II, dementia, emergency surgery, benzodiazepine, and mechanical ventilation use.

### Results

Of the total 200 patients consecutively recruited, 100 were exposed to a no-visitation policy. The number of patients who developed delirium during ICU stay during the before phase and the after phase were 59 (59%) and 64 (64%), respectively (P = 0.127). The adjusted HR of no-visitation policy for the number of days until the first development of delirium during the ICU stay was 0.895 (0.613–1.306).

### Conclusion

The no-visitation policy was not associated with the development of delirium in ICU patients.

**Data Availability Statement:** All relevant data are within the paper and Supporting information files.

**Funding:** TU received Research Grant from Sapporo City University: Grant Number 2021. The funders had no role in study design, data collection

and analysis, decision to publish, or preparation of
the manuscript.

**Competing interests:** The authors have declared
that no competing interests exist.

## Introduction

Delirium is an important problem for critically ill patients, occurring in 83% of mechanically ventilated patients during their intensive care unit (ICU) stay, and in approximately 30% of patients during their ICU stay [1–3]. The development of delirium during ICU stay is associated with longer hospital stays and higher mortality rates [1, 4]. Various interventions have been conducted to prevent delirium in critically ill patients [5, 6]. Among these interventions, visitation has been considered as a possible measure to prevent delirium [7]. The types of visitations in the ICU can be categorized in several ways [8]. In open visitation, visitors are allowed to visit at any time and approximately 30% of the world's ICUs use this method [9]. Next, in restricted visitation, visitors are not allowed to visit except during the hours they are allowed to visit. According to a worldwide survey, approximately 70% of ICUs have restricted visitation [9]. Additionally, in flexible visitation, which was used in past research interventions, visitors were allowed 12 hours of visitation per day [7, 10]. Hospital or ICU-based no-visitation policy is generally not allowed due to ethical and common-sensical reasons, except for the measurement of infection. In Japan, many ICUs have restricted visitation, and 75% of ICUs have a time limit for a single visitation [11].

Previous studies have indicated that flexible visitation policies have been associated with a lower incidence of delirium as compared to restricted visitation policies in ICUs [7, 12]. However, in a recent randomized controlled trial (RCT) that examined the effect of flexible and restricted visitation on the development of delirium, no significant effect of flexible visitation on preventing delirium was confirmed [10]. That is, a longer visitation time was not associated with the development of delirium. However, few studies have examined whether a no-visiting policy influenced the development of delirium. When visitation is prohibited, patients are unable to see their families or loved ones, which may contribute to the development of delirium.

In December 2019, the first coronavirus disease 2019 (COVID-19) case was identified in Wuhan, Hubei Province, China. The COVID-19 has since become a pandemic. In Japan, COVID-19 has repeatedly spread and decreased. To prevent the introduction of severe acute respiratory syndrome coronavirus 2 (SARS-CoV-2) from outside the hospital, some hospitals in Japan have taken measures to prohibit family members from visiting patients.

A previous before-after comparative study examining whether no-visitation policy is associated with the development of delirium in the emergency admission population during the COVID-19 pandemic [13] concluded that a no-visitation policy was associated with a higher incidence of delirium. However, no studies have been conducted with intensive care unit (ICU) patients. Thus, the purpose of this study was to examine the hypothesis that the no visitation policy was associated with a higher incidence of delirium in critically ill patients. The primary outcome was the incidence of delirium during the ICU stay in this study.

## Materials and methods

### Study design

This was a single-center, before-after comparative, retrospective, and observational study. To prevent the transmission of SARS-CoV-2 between patients and visitors, from February 26, 2020, the hospital imposed a no-visitation policy on visitors to hospitalized patients. This study retrospectively compared the development of delirium before and after the implementation of the no-visitation policy.

### Setting

This study was conducted in the medical-surgical ICU of the Kin-ikyo Chuo Hospital in Sapporo City, Japan. The hospital has 450 beds of which 6 are ICU beds.

The data was collected from September 6, 2019 to October 18, 2020. The period before the implementation of the no-visitation policy lasted between September 6, 2019 and February 25, 2020, and the period after the implementation of the no-visitation policy lasted between February 27, 2020 and October 18, 2020.

## Participants

The inclusion criterion for patients in this study was patients aged 18 years or older who stayed in the ICU for more than 48 hours. The exclusion criteria for this study were patients: (1) readmitted to the ICU during the study period; (2) in a constant coma (Richmond Agitation-Sedation Scale score ≤ -4) during the ICU stay, (3) with delirium at the time of ICU admission, (4) with apparent central nervous system disease as revealed by diagnostic imaging, and (5) with difficulty in communicating [14].

The recruitment process is shown in Fig 1. First, patients who were admitted at the time when the no-visitation policy was began to be implemented were excluded because they had been exposed to both visitation policies. Next, patients who were admitted after the implementation of the no-visitation policy were consecutively screened and enrolled until this number reached 100, according to the sample size calculation. Subsequently, patients who were admitted before the implementation of the no-visitation policy were consecutively screened retrospectively from the time of implementation of the no-visitation policy and enrolled until this number reached 100.

## Before the no-visitation policy (before phase)

Before the implementation of the no-visitation policy, the ICU implemented a restricted visitation policy. Visitors were allowed to visit during the following times: (1) 11:00~12:00, (2) 14:30~16:00, and (3) 18:30~20:00. We defined this phase as the "before phase."

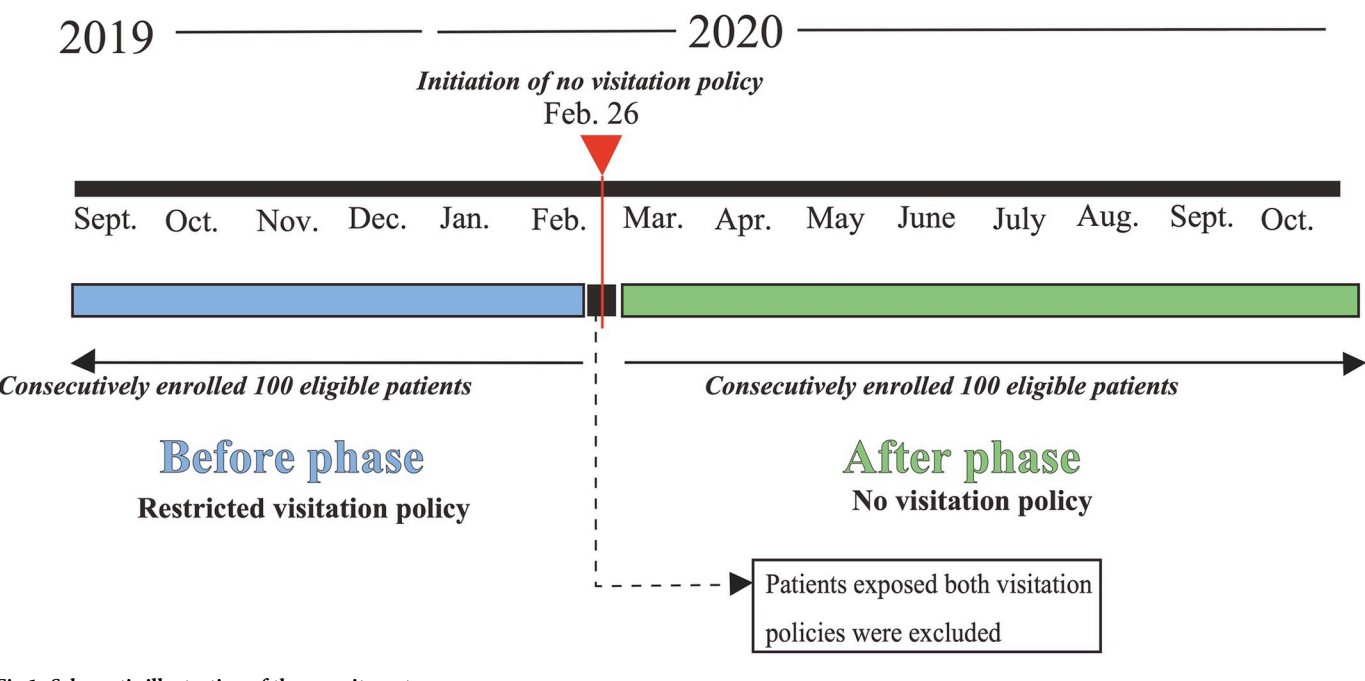

**Fig 1. Schematic illustration of the recruitment process.**

## No-visitation policy in the study (after phase)

No-visitation policy meant that visitors were completely prohibited from visiting the patient while the patient was hospitalized. This phase was defined as "after phase." However, there were a few cases where family members were allowed to visit, such as when the patient's death was inevitable, with the physician or nurse manager's permission. Virtual visiting and telephone communication between patients and their families was uncommon at this time in the ICU. Based on the family's request and equipment availability, such as cell phones, tablets, and smart-phones, we arranged virtual visits or telephone communication between the patient and family.

## Variable/Data collection

We retrospectively collected data from the electronic medical records for this study. We collected patients' characteristics including age, sex, and past medical history, such as dementia and mental disorders. Clinical data, including the primary reason for ICU admission, the Acute Physiology and Chronic Health Evaluation II (APACHE II), and the Sequential Organ Failure Assessment (SOFA) within 24 hours of ICU admission were also collected. Additionally, duration of mechanical ventilation, ICU length of stay, hospital length of stay, opioid use, sedatives use, mortality, use of continuous renal replacement therapy, use of intra-aortic balloon pumping, use of extra corporeal oxygen exchange during ICU stay were recorded. Sepsis was defined as the presence of an infectious disease diagnosis and an elevated SOFA score of two or more points [15]. Benzodiazepines use was defined as continuous intravenous benzodiazepines for more than 24 hours. Mental disorders comprised any mental illness, including schizophrenia and depression. However, dementia was not considered as a mental disorder in this study because we used it as a covariate in the analysis.

To detect delirium, the Intensive Care Delirium Screening Checklist (ICDSC) [16] was obtained from the electronic medical chart. The ICDSC was routinely evaluated three times a day (at 08:00,16:00, and 24:00) by trained ICU nurses and recorded on the medical chart three times a day until patients were discharged from the ICU. The ICDSC is a 0–8-point scale, with a score of four or higher indicating that the patient had delirium [16]. The diagnostic characteristics of ICDSCs were evaluated. The sensitivity and specificity of the Japanese version of the ICDSC with a cutoff of four points were as follows: 97% sensitivity and 97% specificity of the ICDSC for delirium [17]. Additionally, a meta-analysis revealed that the pooled sensitivity and specificity of the ICDSC were 74% (95% CI: 65.3 to 81.5%) and 81.9% (95% CI: 76.7 to 86.4%), respectively [18]. If a patient scored four or more points on the ICDSC at least once during the day, the patient was considered to have delirium in this study. Visitation for patients was noted in the medical records, and we extracted the data throughout the study period.

## Sample size

To analyze the association between the number of days until the development of delirium and visitation policies, the sample size was calculated as follows: the incidence of delirium as an event was assumed to be 40% based on previous literature [1–3]. The covariates to be adjusted in the Cox proportional-hazards model were set to 7. Based on these conditions, we used the rule that 10 events per variable were required to perform a Cox proportional-hazards regression and set the required sample size to 200 [19, 20].

## Statistical analysis

The data obtained were expressed as median and interquartile range (IQR) for continuous variables and as proportions for categorical variables. Fisher's exact, $\chi$-square, and Wilcoxon rank-sum tests were used to compare patient characteristics in the two phases.

The patients were divided into two groups, before and after the implementation of the no-visitation policy, and were compared using Kaplan-Meier curves with the number of days until the development of delirium as the outcome. The log-rank test was used to compare the two groups. In the multivariate analysis, we used the Cox proportional-hazards models to analyze the association between the number of days until the onset of delirium as the objective variable and the visitation policies. We calculated the hazard ratio (HR) and 95% confidence interval (CI) for the development of delirium. We predefined covariates for multivariable analysis based on past studies and clinical expert knowledge. We selected age [21–23], sex, APACHE II [21], history of dementia [24], ICU admission after emergency surgery [22, 23], use of benzodiazepine [21, 23, 25], and mechanical ventilation use [26] for covariates. To avoid multicollinearity, the APACHE II score was calculated without using age as a covariate. Dose of opioids was not included in the model, because most patients receiving opioids were also receiving mechanical ventilation.

Simple imputation was used to handle missing values [27] for the ICDSC. We defined deficit as the day when none of the three evaluations were recorded. The missing values of the ICDSC were replaced by the average of the day before or after. For other missing variables, we planned to use multiple imputations.

All statistical tests were two-tailed, and the statistical significance was set at 0.05. We used R statistical software (version 4.0.2, The R Foundation for Statistical Computing, Vienna, Austria, https://www.r-project.org/) for the statistical analysis.

## Sensitivity analysis

We conducted a sensitivity analysis to test the robustness of the model's assumptions. We conducted three sensitivity analyses.

First, we conducted a multivariate logistic regression analysis and analyzed the relationship between the development of delirium and the visitation policies. The same covariates were included in this model for multivariable adjustment.

Second, in order to remove the effects of duration of mechanical ventilation on our findings, we changed the covariate of mechanical ventilation use as binomial variable to the duration of mechanical ventilation, and conducted the Cox proportional-hazards model. All covariates included in the primary analysis were also included into the model, with the exception of mechanical ventilation use.

Third, a previous study reported the occurrence of false positives when the ICDSC was used to determine delirium in patients with dementia and mental disorders [16]. Therefore, we conducted an analysis using Cox proportional-hazards model, excluding patients with dementia or mental disorders. All covariates were the same as the ones in the primary analysis except for dementia.

## Ethical considerations

Ethical approval for the research protocol was granted by the ethical review board of the Graduate School of Nursing, Sapporo City University, Sapporo, Japan, (approval ID 2021-9-3) and the participating institution (approval ID 2020–44). The requirement for informed consent was waived because of the anonymous nature of the data.

## Results

### Participant characteristics

The patient flowchart is shown in Fig 2. Throughout the study period, 467 patients were admitted to the ICU. A total of 267 patients were excluded. Consequently, a total of 200

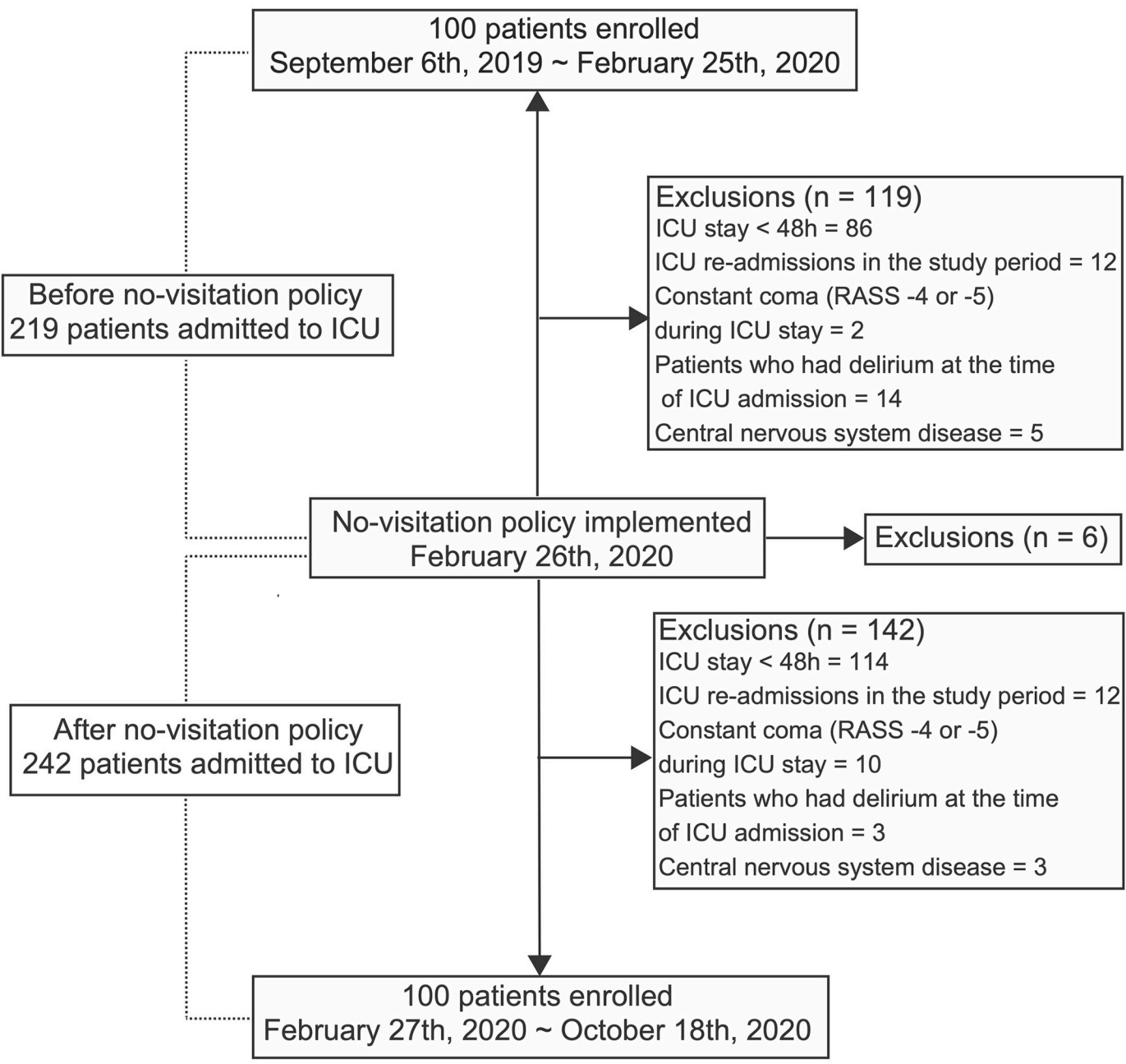

**Fig 2. Patient recruitment flowchart.**

patients (100 patients from the before phase and 100 from the after phase) were included in this study (Fig 2). Patient characteristics are shown in Table 1. Patients from the after phase had significantly higher APACHE II scores, mortality rates, mechanical ventilator use rates, and longer duration of mechanical ventilation, compared to those from the before phase. No patients with COVID-19 were admitted in the ICU during the study period.

Deficits in ICDSC data were observed in 2.4% of the patients and were only limited to the day of ICU admission or the day of ICU discharge. Therefore, missing values on the day of ICU admission were replaced by the average ICDSC score of the following day, and deficits on

**Table 1. Patient characteristics before and after the no-visitation policy.**

| Variables | n = 200 | Before phase n = 100 | After phase n = 100 | p-value[a] |
|---|---|---|---|---|
| Age, median [IQR] | 76.0 [68.8–84.2] | 75.5 [67.8–84.2] | 76.0 [69.0–84.2] | .841 |
| Age ≧ 65 yr, n (%) | 161 (80.5) | 80 (80.0) | 81 (81.0) | .858 |
| Female, n (%) | 76 (38.0) | 40 (40.0) | 36 (36.0) | .560 |
| **Pre-existing medical condition** | | | | |
| Hypertension, n (%) | 103 (51.5) | 45 (45.0) | 58 (58.0) | .066 |
| Dementia, n (%) | 28 (14.0) | 10 (10.0) | 18 (18.0) | .103 |
| Mental disorder[b], n (%) | 21 (10.5) | 12 (12.0) | 9 (9.0) | .489 |
| **ICU admission type, n (%)** | | | | |
| Medical | 144 (72.0) | 68 (68.0) | 76 (76.0) | .367 |
| Emergency surgery | 19 (9.5) | 12 (12.0) | 7 (7.0) | |
| Scheduled surgery | 37 (18.5) | 20 (20.0) | 17 (17.0) | |
| **Primary reason for ICU admission, n (%)** | | | | |
| Respiratory failure | 53 (26.5) | 30 (30.0) | 23 (23.0) | .055 |
| CHF/ACS/Arrhythmia | 28 (14.0) | 7 (7.0) | 21 (21.0) | |
| Sepsis | 28 (14.0) | 15 (15.0) | 13 (13.0) | |
| Cardiovascular surgery | 25 (12.5) | 16 (16.0) | 9 (9.0) | |
| Abdominal surgery | 15 (7.5) | 9 (9.0) | 6 (6.0) | |
| Other surgery | 15 (7.5) | 5 (5.0) | 10 (10.0) | |
| Others | 36 (18.0) | 18 (18.0) | 18 (18.0) | |
| ICU length of stay, median [IQR] | 5.0 [4.0–8.2] | 5.0 [4.0–7.2] | 6.0 [4.0–10.0] | .209 |
| Hospital length of stay, median [IQR] | 24.0 [13.8–42.0] | 22.5 [13.0–42.0] | 25.5 [15.0–39.8] | .166 |
| APACHE II, median [IQR] | 18.0 [12.0–23.2] | 17.0 [11.0–22.0] | 19.5 [13.8–26.2] | .008 |
| SOFA, median [IQR] | 6.0 [4.0–9.0] | 6.0 [3.8–9.0] | 7.0 [4.0–9.0] | .105 |
| Mechanical ventilation use, n (%) | 117 (58.5) | 50 (50.0) | 67 (67.0) | .015 |
| Duration of mechanical ventilation, median [IQR] | 4.0 [2.0–8.0] | 3.0 [2.0–6.8] | 5.0 [3.0–10.0] | .013 |
| Opioid use, n (%) | 114 (57.0) | 46 (46.0) | 68 (68.0) | .002 |
| Benzodiazepine use, n (%) | 16 (8.0) | 7 (7.0) | 9 (9.0) | .602 |
| Sedative use, n (%) | 134 (67.0) | 63 (63.0) | 71 (71.0) | .229 |
| CRRT, n (%) | 28 (14.0) | 11 (11.0) | 17 (17.0) | .221 |
| IABP, n (%) | 10 (5.0) | 5 (5.0) | 5 (5.0) | >.999 |
| ECMO, n (%) | 5 (2.5) | 2 (2.0) | 3 (3.0) | >.999 |
| **Mortality, n (%)** | | | | |
| Death in hospital, n(%) | 27 (13.5) | 7 (7.0) | 20 (20.0) | .026 |
| Death in ICU, n (%) | 18 (9.0) | 10 (10.0) | 8 (8.0) | |

[a]Wilcoxon rank sum test; Pearson's chi-squared test; Fisher's exact test.

[b]Mental disorder; schizophrenia, depression, bipolar disorder, alcoholism, adjustment disorder, and panic disorder.

IQR, interquartile range; CHF/ACS, congestive heart failure/acute coronary syndrome; APACHE II, Acute Physiology and Chronic Health Evaluation II; SOFA, sepsis-related organ failure assessment; CRRT, continuous renal replacement therapy; IABP, intra-aortic balloon pumping; ECMO, extracorporeal membrane oxygenation.

the day of ICU discharge were replaced by the average of the ICDSC score of the previous day. There were no missing data other than the ICDSC scores mentioned above.

## Univariable analysis of outcome data

Of the 200 patients admitted to the ICU during the study period, 123 (61.5%) patients developed delirium. The difference in the incidence of delirium between the before and after phases (59% vs. 64%, $p = 0.127$) was not statistically significant. The number of visits per patient

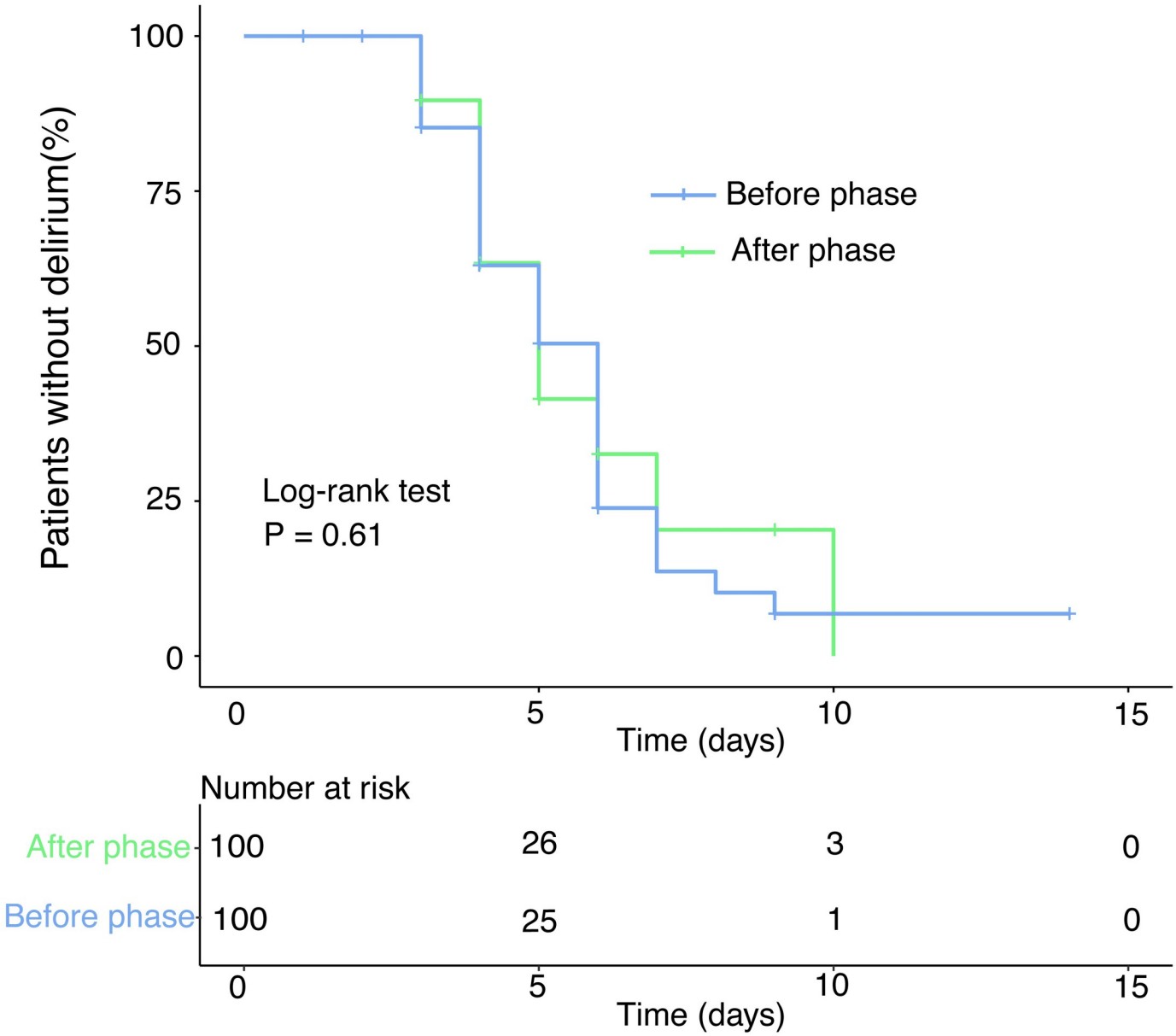

**Fig 3. Kaplan-Meier curve: Time until the development of delirium during the ICU stay for the before and after phases.**

during ICU stay in each phase is showed in S1 Fig. The proportion of patients who were visited during the after phase was significantly lower than that during the before phase (19% vs. 92%, p<0.001).

The median number of days for onset of delirium among patients who developed delirium before and after the implementation of the no-visitation policy was not significantly different (2 [IQR], 2–3 vs 2 [IQR], 2–3, $p = 0.696$). This result does not take into account whether patients had been sedated or not. The Kaplan-Meier curve between the before and after phases for the development of delirium is shown in Fig 3. The log-rank test did not show a significant difference between the two phases ($p = 0.61$).

**Table 2. Estimates of the hazard ratios of variables on development of delirium in the Cox proportional-hazards models.**

| Variable | unadjusted | | adjusted[a] | |
|---|---|---|---|---|
| | HR (95%CI) | p-value | HR (95%CI) | p-value |
| No-visitation policy | 1.157 (0.810–1.652) | .421 | 0.895 (0.613–1.306) | .565 |
| Age | 1.004 (0.991–1.017) | .543 | 0.998 (0.984–1.012) | .778 |
| Male | 1.109 (0.765–1.607) | .585 | 1.118 (0.757–1.653) | .573 |
| Dementia | 2.176 (1.383–3.421) | < .001 | 2.078 (1.251–3.454) | .004 |
| Emergency surgery | 1.705 (0.990–2.935) | .054 | 1.512 (0.831–2.750) | .174 |
| APACHE II[b] | 1.048 (1.026–1.071) | < .001 | 1.039 (1.015–1.064) | .001 |
| Benzodiazepine use | 1.484 (0.849–2.593) | .166 | 1.071 (0.592–1.939) | .819 |
| Mechanical ventilation use | 2.317 (1.550–3.463) | < .001 | 1.735 (1.100–2.736) | .017 |

[a]The Cox proportional-hazards model was used to adjust for eight variables: no-visitation policy, age, sex, dementia, emergency surgery, APACHEII, benzodiazepine use, and mechanical ventilation use.

[b]APACHEII score was calculated without age related score.

HR, hazard ratio; CI, Confidence interval.

## Multivariable analysis

The results of the unadjusted and adjusted Cox regression models are presented in Table 2. The no-visitation policy was not significantly associated with the development of delirium after adjusting for covariates (HR 0.895, 95% CI, 0.613–1.306; $p$ = 0.525). The presence of dementia, higher APACHE II score, and mechanical ventilation use were significantly associated with the development of delirium.

## Sensitivity analysis

First, the multivariate logistic regression was performed to determine the outcome of the development of delirium and its results are presented in S1 Table. The results indicated that the no-visitation policy was not significantly associated with the development of delirium (OR 0.714, 95%CI 0.354–1.415). Second, the results of the Cox proportional-hazards model with a covariate changed from mechanical ventilation use to duration of mechanical ventilation are shown in S2 Table. The results showed no association between the no-visitation policy and the number of days until the development of delirium (HR 0.938, 95%CI 0.644–1.367). Third, the results of the Cox proportional-hazards model without the inclusion of patients with dementia or mental disorders are shown in S3 Table. The results showed no association between the no-visitation policy and the number of days until the development of delirium (HR 0.890, 95%CI 0.572–1.385).

## Discussion

In this single-center before-after, comparative, and retrospective observational study, the no-visitation policy was not associated with the development of delirium in critically ill patients. The robustness of the findings is demonstrated in the sensitivity analysis.

Based on our findings, no visitation may not be associated with delirium in critically ill patients. The results of this study are consistent with the results of a previous RCT comparing the incidence of delirium in flexible visitation and restricted visitation [10]. However, in a previous observational before-after study comparing no visitation and restricted visitation in emergency patients, the incidence of delirium was higher in the no-visitation group [13]. There are several possible reasons for these different results. In the present study, we screened

delirium three times a day using ICDSC by trained nurses; however, past studies did not screen every patient. They considered patients to be diagnosed with delirium by a psychiatrist based on consultation. As noted, detection bias may not be avoided during the COVID-19 pandemic; however, the number of consultations before and after visitation policy change has not been reported. In addition, the targeted patient population was different. In this study, critically ill patients were targeted, while in the previous study, emergency inpatients were targeted.

There are several possible reasons for the lack of association between the number of days until the development of delirium and the no visitation policy. First, there are many risk factors for delirium in critically ill patients. In the ICU, risk factors for delirium include drugs, severity of illness, emergency surgery, and invasive treatments such as mechanical ventilators [28]. Therefore, even if visitation is effective in preventing delirium, it may not lead to the prevention of delirium in ICU patients, given the iatrogenic factors that contribute to the development of delirium. This is probably the main reason for finding no relationship between the number of days until the development of delirium and the no visitation policy in this study. Second, the impaired level of consciousness of ICU patients may have contributed to the lack of association between the number of days until the development of delirium and the no visitation policy. In this study, 58% of the patients were on mechanical ventilators, and 67% were receiving sedatives. Therefore, it is likely that more than half of the patients had a period of poor consciousness. The incidence of delirium has been reduced in a before-after comparative study comparing restricted visitation with flexible visitation [7]. However, as compared to the patients in our study, the proportion of patients with mechanical ventilation use and sedative use in the previous study was less than half. The patients may not recognize the visitors as a result of poor consciousness. This may reduce the effectiveness of visits to prevent delirium.

## Strengths of the study

This study has two strengths: First, it investigated the association between no-visitation policy and delirium in critically ill patients. No visitation cannot be implemented as an intervention due to ethical issues. Therefore, a study like this would not have been possible under normal circumstances and could only be conducted under special circumstances. Second, no-visitation policy was implemented as a measure against infectious diseases and not for the purpose of this study; hence, there was no performer bias.

## Limitations of the study

This study has several limitations: First, as it was a before-and-after comparative study, unknown confounding factors may not have been controlled for. Second, this study was a single-center study; thus, external validity should be considered with caution. Third, even with a no-visitation policy, a few visitors stayed with the patients for a short time (i.e., a brief period of time just before the patient's death); however, we evaluated the no-visitation policy, not no-visitation. Thus, we did not exclude the patients who met visitors. Given the paucity of visitation opportunities and the short time, we did not consider that visitation during the after period interfered with our findings. Fourth, there may be an Information bias regarding the presence of dementia or mental disorders of the variables used in the multivariate and sensitivity analyses. The presence of these diseases was determined by the medical records, which may have contributed to a bias because there might have been cases that were either not disclosed in the medical records or were documented as mild mental disorders. However, we considered that those cases occurred randomly in both phases and did not significantly affect the results. Fifth, the exact number of online communications between patients and family members

during the no-visitation policy is unknown. However, we did not have online communication manual and structured, thus we considered the contribution of online communication on our findings to have been minimum.

## Implications for clinical practice

We emphasize that the results of this study do not indicate that ICU patients do not need to have visitors. This is because delirium is only one indicator of the effectiveness of ICU visits. A previous study showed that flexible family visitation to critically ill patients reduce family anxiety and depression and increases satisfaction [10]. Another study showed a reduction in patients' own anxiety symptoms under unrestricted visitation [29]. Therefore, visitation has the potential for a variety of positive effects on critically ill patients and their families. Above all, visitation is the right of patients and their families. However, the prohibition of visitation was not associated with the development of delirium, which is important for clarifying the mechanism of delirium.

## Conclusion

This study showed that no visitation policy was not associated with the incidence of delirium and the number of days until the development of delirium in critically ill patients in the ICU.

## Supporting information

**S1 Fig. Histograms of the number of visits per patient during ICU stay in each phase.**
(TIF)

**S1 Table. Estimates of the adjusted odds ratios of variables on the incidence of delirium in the multivariate logistic regression.**
(DOCX)

**S2 Table. Estimates of the adjusted hazard ratios of variables on the development of delirium in the Cox proportional-hazards models.**
(DOCX)

**S3 Table. Estimates of the adjusted hazard ratios of variables on the development of delirium in the Cox proportional-hazards models removing patients with dementia or mental disorders from the primary analysis.**
(DOCX)

**S1 Data. Dataset used for Kaplan-Meier curve showing time to delirium during ICU stay in each phase and used for histogram showing visitors per patient in each phase.**
(XLSX)

## Acknowledgments

We would like to thank the staff at Kin-ikyo Chuo Hospital who assisted the authors in this study.

## Author Contributions

**Conceptualization:** Fumihide Shinohara, Takeshi Unoki.

**Data curation:** Fumihide Shinohara.

**Formal analysis:** Fumihide Shinohara.

**Funding acquisition:** Takeshi Unoki.

**Investigation:** Fumihide Shinohara, Megumi Horikawa.

**Methodology:** Fumihide Shinohara, Takeshi Unoki.

**Project administration:** Fumihide Shinohara, Takeshi Unoki.

**Supervision:** Takeshi Unoki.

**Validation:** Fumihide Shinohara, Takeshi Unoki.

**Visualization:** Fumihide Shinohara.

**Writing – original draft:** Fumihide Shinohara.

**Writing – review & editing:** Takeshi Unoki, Megumi Horikawa.

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
