## [Decision Letter · Decision Letter 0]

29 Dec 2021

PONE-D-21-25576Effects of no-visitation policy on the development of delirium in patients admitted to the intensive care unitPLOS ONE

Dear Dr. Unoki,

Thank you for submitting your manuscript to PLOS ONE. After careful consideration, we feel that it has merit but does not fully meet PLOS ONE’s publication criteria as it currently stands. Therefore, we invite you to submit a revised version of the manuscript that addresses the points raised during the review process.

We look forward to receiving your revised manuscript.

Kind regards,

Andrea Ballotta

Academic Editor

PLOS ONE

Journal Requirements:

2. This is a retrospective study , as such, we do not feel that any conclusions on the intervention effects can be supported; thus, we ask that you revise the text (especially, but not limited to, the title) to avoid unsupported statements.

Additional Editor Comments (if provided):

I apologize for the delay .

On the basis of the reviewers’ comments the manuscript needs major revision

Reviewers' comments:

Reviewer's Responses to Questions

**Comments to the Author**

1. Is the manuscript technically sound, and do the data support the conclusions?

Reviewer #1: Partly

Reviewer #2: Partly

2. Has the statistical analysis been performed appropriately and rigorously? 

Reviewer #1: No

Reviewer #2: Yes

3. Have the authors made all data underlying the findings in their manuscript fully available?

Reviewer #1: Yes

Reviewer #2: Yes

4. Is the manuscript presented in an intelligible fashion and written in standard English?

Reviewer #1: Yes

Reviewer #2: Yes

5. Review Comments to the Author

Reviewer #1: In this study, the authors investigated the association between prohibited visitation policies and delirium in 200 intensive care unit (ICU) patients. They found that the adjusted HR of no-visitation policy for the number of days until the first development of delirium during the ICU stay was 0.883 (0.603–1.294). So they concluded that the no-visiting policy was not associated with development of delirium in ICU patients. It is an interesting study, focusing on an important issue and it provides informative conclusions. I offer the following comments:

It is not clear the study design. It seems that it is a case-control study rather than a before-after comparative study. 200 patients were enrolled: 100 before and 100 after; so it seems that the enrollment was not consecutive. Please, better clarify this point.

The paragraph “Bias” needs to be better explained. The authors quote: “Differences in characteristics before and after the implementation of the 168 no-visitation policy existed. Therefore, we used multivariate analysis to adjust 169 for known confounders associated with delirium”. This point should be clearly explained and better addressed in the statistical analysis paragraph and in the results section. Moreover, the authors report in the Statistical analysis: “The covariates were as follows: age, sex, APACHE-II, history of dementia, ICU, admission after emergency surgery, use of benzodiazepine, and use of 194 mechanical ventilation. To avoid multicollinearity, the APACHE II score was 195 calculated without using age as a covariate”. How were these variables chosen (step-wise selection? Univariate analysis”? This point is critical for the analysis. All these points might have influenced study results.

The paragraph “Sample size” is not clear and the sample size calculation should be better reported and more robustly based on statistical data (power of the analysis?)

Lines 249-251: The incidence of delirium was higher during 250 the no-visitation policy than during the usual visitation policy (64% vs. 59%, 251 p=0.127). This sentence should be rephrased as there is no difference in the incidence of delirium between the two groups.

Table 2. It is not clear whether these variables for adjusted for each others. Please, specify.

Figure is interesting but it should be improved from a graphical point of view.

Reviewer #2: Takeshi Unoki et al. Present a retrospective study on the impact of the no-visitation policy on patients admitted to ICU during Covid-19 pandemic. They concluded that the no visitation policy was not associated

to an higher incidence of delirium.

Anyway, I do have some comments.

1) The calculation of the sample size should be better defined

2) In the bias section, the authors declare to use multivariate analysis to adjust for known

confounders associated with delirium. Can the authors please state which are the

counfounders and how they were chosen?

3) In the “No visitation policy in the study” paragraph, the authors state that some patients had

access to online communications with the families. How many patients had the possibility? Do

you think this might be a bias?

4) Can the authors please explain what they mean with “medical ventilation”?

5) Can the authors define which were the psychiatric disorders taken into account (i.e.

depression/schizophrenia etc)?

6) How did the authors address the patients with dementia or psychiatric disorders in the ICDSC

scale? I wonder if those patients should be excluded from the analysis.

7) I wonder what is the real impact of the higher percentage of opioids use and the higher number

of mechanical ventilation days in the no visitation group. As the authors state it as a study

limitation, could it be possible to perform a subanalysis excluding such a difference?

8) In line 252-253, the authors talk about the median number of days for onset of delirium.

Among these days, do the authors take into account also the days the patients were sedated?

6. PLOS authors have the option to publish the peer review history of their article (what does this mean?). If published, this will include your full peer review and any attached files.

Reviewer #1: No

Reviewer #2: No

---

## [Author Response · Author response to Decision Letter 0]

3 Feb 2022

Response to the Editor and Reviewers

Comment

Journal Requirements:

Response

We have revised the description of the manuscript according to PLOS ONE's style requirements.

Comment

2. This is a retrospective study, as such, we do not feel that any conclusions on the intervention effects can be supported; thus, we ask that you revise the text (especially, but not limited to, the title) to avoid unsupported statements.

Response

We totally agree with your comments. We have revised the title and relooked through the entire manuscript to ensure that any terms indicating the intervention effects were not used.

 ------Revised Manuscript------

Modified title: Relationship between no-visitation policy and the development of delirium in patients admitted to the intensive care unit

Comment

Response 

We have added the grant number for this study in the ‘Funding Information’ section.

Comment

Reply

We have submitted the required minimum data set as S1 Data.

Reviewer’s comments:

Reviewer’s Responses to Questions

Comment to the Author

Comment

Reviewer #1:

In this study, the authors investigated the association between prohibited visitation policies and delirium in 200 intensive care unit (ICU) patients. They found that the adjusted HR of no-visitation policy for the number of days until the first development of delirium during the ICU stay was 0.883 (0.603–1.294). So they concluded that the no-visiting policy was not associated with development of delirium in ICU patients. It is an interesting study, focusing on an important issue and it provides informative conclusions. I offer the following comments:

1. It is not clear the study design. It seems that it is a case-control study rather than a before-after comparative study. 200 patients were enrolled: 100 before and 100 after; so it seems that the enrollment was not consecutive. Please, better clarify this point.

Reply

We apologize for the confusion due to our disorganized description. We revised the description of the methods and added a new Fig1 that explains the study design to make it easier easily to understand the methods. 

This study is not a case-controlled study, but a single-center before-after, comparative, and retrospective observational study. Before and after the implementation of the no-visitation policy, eligible patients were consecutively enrolled until their number reached 100, respectively. We have revised the description of the methods and Fig.2 and added a new Fig 1. The revised description in the manuscript is as below, with the changed parts in red. 

 ------Revised Manuscript------

P8, Line117

The recruitment process is shown in Fig 1. First, patients who were admitted at the time when the no-visitation policy was began to be implemented were excluded because they had been exposed to both visitation policies. Next, patients who were admitted after the implementation of the no-visitation policy were consecutively screened and enrolled until this number reached 100, according to the sample size calculation. Subsequently, patients who were admitted before the implementation of the no-visitation policy were consecutively screened retrospectively from the time of implementation of the no-visitation policy and enrolled until this number reached 100.

Comment

2. The paragraph “Bias” needs to be better explained. The authors quote: “Differences in characteristics before and after the implementation of the 168 no-visitation policy existed. Therefore, we used multivariate analysis to adjust 169 for known confounders associated with delirium”. This point should be clearly explained and better addressed in the statistical analysis paragraph and in the results section. Moreover, the authors report in the Statistical analysis: “The covariates were as follows: age, sex, APACHE-II, history of dementia, ICU, admission after emergency surgery, use of benzodiazepine, and use of 194 mechanical ventilation. To avoid multicollinearity, the APACHE II score was 195 calculated without using age as a covariate”. How were these variables chosen (step-wise selection? Univariate analysis”? This point is critical for the analysis. All these points might have influenced study results.

Reply

We apologize for the confusion caused due to inappropriate wording. 

We attempted to explain how we analyzed and how we choose the covariates. First, the bias section was removed and a description regarding the analysis was integrated into the “Statistical analysis” section. 

We choose the covariates based on past research that described factors for delirium incidents in ICU settings and clinical expert knowledge. This method was recommended, whereas the selection of covariates based on statistical methods (i.e., stepwise method) was not recommended by the guideline (Lederer et al., 2019).

Reference

Lederer, D. J., Bell, S. C., Branson, R. D., Chalmers, J. D., Marshall, R., Maslove, D. M., Ost, D. E., Punjabi, N. M., Schatz, M., Smyth, A. R., Stewart, P. W., Suissa, S., Adjei, A. A., Akdis, C. A., Azoulay, É., Bakker, J., Ballas, Z. K., Bardin, P. G., Barreiro, E., … Vincent, J.-L. (2019). Control of confounding and reporting of results in causal inference studies. Guidance for authors from editors of respiratory, sleep, and critical care journals. Annals of the American Thoracic Society, 16(1), 22–28. doi: 10.1513/AnnalsATS.201808-564PS

------Revised Manuscript------

P11, Line 188

The patients were divided into two groups, before and after the implementation of the no-visitation policy, and were compared using Kaplan-Meier curves with the number of days until the development of delirium as the outcome. The log-rank test was used to compare the two groups. In the multivariate analysis, we used the Cox proportional-hazards models to analyze the association between the number of days until the onset of delirium as the objective variable and the visitation policies. We calculated the hazard ratio (HR) and 95% confidence interval (CI) for the development of delirium. We predefined covariates for multivariable analysis, based on past studies and clinical expert knowledge. We selected age [21-23], sex, APACHE-II [21], history of dementia [24], ICU admission after emergency surgery [22,23] , use of benzodiazepine [21,23,25] and mechanical ventilation use [26] for covariates. To avoid multicollinearity, the APACHE II score was calculated without using age as a covariate. Dose of opioids was not included in the model, because most patients receiving opioids were also receiving mechanical ventilation. 

Comment

3. The paragraph “Sample size” is not clear and the sample size calculation should be better reported and more robustly based on statistical data (power of the analysis?)

Reply

We apologize that the description was incomplete and contained errors.

In fact, when we calculated the sample size, we used the rule that 10 is the required number of events per variable in a Cox proportional-hazards regression analysis (Peduzzi et al., 1995). We also pre-determined seven variables to be adjusted in the Cox proportional-hazards model to examine the association between the no visitation policy and delirium. In addition, based on the previous literature, we estimated the incidence of delirium was around 40%. Based on this assumption, a sample size of 200 was required. We revised the sample size section. The revised manuscript is as below, with the changed parts in red.

Reference

Peduzzi, P., Concato, J., Feinstein, A. R., & Holford, T. R. (1995). Importance of events per independent variable in proportional hazards regression analysis. II. Accuracy and precision of regression estimates. Journal of Clinical Epidemiology, 48(12), 1503–1510. doi:

10.1016/0895-4356(95)00048-8

------Revised Manuscript------

P10 Line:174

Sample size

To analyze the association between the number of days until the development of delirium and visitation policies, the sample size was calculated as follows: the incidence of delirium as an event was assumed to be 40% based on previous literature [1–3]. The covariates to be adjusted in the Cox proportional-hazards model were set to 7. Based on these conditions, we used the rule that 10 events per variable are required to perform a Cox proportional-hazards regression and set the required sample size to 200 [19,20].

Comment

4.Lines 249-251: The incidence of delirium was higher during 250 the no-visitation policy than during the usual visitation policy (64% vs. 59%, 251 p=0.127). This sentence should be rephrased as there is no difference in the incidence of delirium between the two groups.

Reply

We totally agree with your comments and have revised the sentence accordingly. The revised description is below, with the changed parts in red.

------Revised Manuscript------

P18 Line:263

Univariable analysis of outcome data 

Of the 200 patients admitted to the ICU during the study period, 123 (61.5%) patients developed delirium. The difference in the incidence of delirium between the before and after phases (59% vs. 64%, p=0.127) was not statistically significant.

Comment

5. Table 2. It is not clear whether these variables for adjusted for each others. Please, specify.

Reply

Thank you for pointing this out.

All covariates described in the statistical section were included in the multivariable analysis for multivariable adjustment. We have shown the variables used for adjustments in the footnote of Table 2. 

------Revised Manuscript------

P20 Line:290

aThe Cox proportional-hazards model was used to adjust for eight variables: no-visitation policy, age, sex, dementia, emergency surgery, APACHEⅡ, benzodiazepine use, and mechanical ventilation use.

Comment

6.Figure is interesting but it should be improved from a graphical point of view.

Reply

Thank you for pointing this out.

The three figures have been completely revised to make them easier to understand. 

Comment

Reviewer #2: Takeshi Unoki et al. Present a retrospective study on the impact of the no-visitation policy on patients admitted to ICU during Covid-19 pandemic. They concluded that the no visitation policy was not associated to an higher incidence of delirium. Anyway, I do have some comments.

1.The calculation of the sample size should be better defined

Reply

We apologize that the description was incomplete and contained errors.

In fact, when we calculated the sample size, we used the rule that 10 is the required number of events per variable in a Cox proportional-hazards regression analysis (Peduzzi et al., 1995). We also pre-determined seven variables to be adjusted in the Cox proportional-hazards model to examine the association between the no visitation policy and delirium. In addition, based on the previous literature, we estimated the incidence of delirium was around 40%. Based on this assumption, a sample size of 200 was required. We revised the sample size section. The revised manuscript is below, with the changed parts in red.

Reference

Peduzzi, P., Concato, J., Feinstein, A. R., & Holford, T. R. (1995). Importance of events per independent variable in proportional hazards regression analysis. II. Accuracy and precision of regression estimates. Journal of Clinical Epidemiology, 48(12), 1503–1510. doi:

10.1016/0895-4356(95)00048-8

------Revised Manuscript------

P10 Line:174

Sample size

To analyze the association between the number of days until the development of delirium and visitation policies, the sample size was calculated as follows: the incidence of delirium as an event was assumed to be 40% based on previous literature [1–3]. The covariates to be adjusted in the Cox proportional-hazards model were set to 7. Based on these conditions, we used the rule that 10 events per variable are required to perform a Cox proportional-hazards regression and set the required sample size to 200 [19,20].

Comment

2. In the bias section, the authors declare to use multivariate analysis to adjust for known confounders associated with delirium. Can the authors please state which are the counfounders and how they were chosen?

Reply

We apologize for the confusion caused by our inappropriate wording. 

We attempted to explain how we analyzed and chose the covariates. First, the bias section was removed and a description regarding the analysis was integrated into the “Statistical analysis” section. Second, we choose the covariates based on past research that described factors for delirium incidents in the ICU settings and clinical expert knowledge. This method was recommended by the guideline (Lederer et al., 2019). We attached a rationale for the research regarding the factors involved in the development of delirium to each of the covariates in the statistical analysis section. We show below the revised manuscript with the modified parts in red.

Reference

Lederer, D. J., Bell, S. C., Branson, R. D., Chalmers, J. D., Marshall, R., Maslove, D. M., Ost, D. E., Punjabi, N. M., Schatz, M., Smyth, A. R., Stewart, P. W., Suissa, S., Adjei, A. A., Akdis, C. A., Azoulay, É., Bakker, J., Ballas, Z. K., Bardin, P. G., Barreiro, E., … Vincent, J.-L. (2019). Control of confounding and reporting of results in causal inference studies. Guidance for authors from editors of respiratory, sleep, and critical care journals. Annals of the American Thoracic Society, 16(1), 22–28. doi: 10.1513/AnnalsATS.201808-564PS

------Revised Manuscript------

P11, Line 188

The patients were divided into two groups, before and after the implementation of the no-visitation policy, and were compared using Kaplan-Meier curves with the number of days until the development of delirium as the outcome. The log-rank test was used to compare the two groups. In the multivariate analysis, we used the Cox proportional-hazards models to analyze the association between the number of days until the onset of delirium as the objective variable and the visitation policies. We calculated the hazard ratio (HR) and 95% confidence interval (CI) for the development of delirium. We predefined covariates for multivariable analysis, based on past studies and clinical expert knowledge. We selected age [21-23], sex, APACHE-II [21], history of dementia [24], ICU admission after emergency surgery [22,23] , use of benzodiazepine [21,23,25] and mechanical ventilation use [26] for covariates. To avoid multicollinearity, the APACHE II score was calculated without using age as a covariate. Dose of opioids was not included in the model, because most patients receiving opioids were also receiving mechanical ventilation. 

-------------------

Comment

3. In the “No visitation policy in the study” paragraph, the authors state that some patients had access to online communications with the families. How many patients had the possibility? Do you think this might be a bias?

Reply

Thank you for pointing this out.

Yes, we think bias may occur due to online communication in the no-visitation phase, as you indicated. However, we do not considered that this bias significantly affected our results. Firstly, there were no manuals, tablets, or other equipment for online communication at the hospital during the study period. We considered that this indicated that online communication was not frequent. Of course, If a family member brought a cell phone to the hospital and the patient could use it, they could call each other. Unfortunately, online communication was not recorded in the medical chart; therefore, we were not able to be quantify it. We have described the potential bias regarding online communication in the section on limitations of the study. We show below the revised manuscript with the modified parts in red.

------Revised Manuscript------

P24, Line 359

Limitations of the study

This study has several limitations: First, as it was a before-and-after comparative study, unknown confounding factors may not have been controlled for. Second, this study was a single-center study; thus, external validity should be considered with caution. Third, even with a no-visitation policy, a few visitors stayed with the patients for a short time (i.e., a brief period of time just before the patient’s death); however, we evaluated the no-visitation policy, not no-visitation. Thus, we did not exclude the patients who met visitors. Given the paucity of visitation opportunities and the short time, we did not consider that visitation during the after period interfered with our findings. Fourth, there may be an Information bias regarding the presence of dementia or mental disorders of the variables used in the multivariate and sensitivity analyses. The presence of these diseases was determined by the medical records, which may have contributed to a bias because there might have been cases that were either not disclosed in the medical records or were documented as mild mental disorders. However, we considered that those cases occurred randomly in both phases and did not significantly affect the results. Fifth, the exact number of online communications between patients and family members during the no-visitation policy is unknown. However, we did not have online communication manual and structured, thus we considered the contribution of online communication on our findings to have been minimum. 

Comment

4. Can the authors please explain what they mean with “medical ventilation”?

Reply

We apologize for the typo. The correct term is “mechanical ventilation use.” We have corrected it throughout the manuscript.

Comment

5. Can the authors define which were the psychiatric disorders taken into account (i.e.depression/schizophrenia etc)?

Reply

Thank you for pointing this out.

We added the description of the definition of the mental disorders in this study. We show below the revised manuscript with the modified parts in red.

------Revised Manuscript------

P9 Line 144

Variable / Data collection

We retrospectively collected data from the electronic medical records for this study. We collected patients’ characteristics including age, sex, and past medical history, such as dementia and mental disorders. Clinical data, including the primary reason for ICU admission, the Acute Physiology and Chronic Health Evaluation II (APACHE II), and the Sequential Organ Failure Assessment (SOFA) within 24 hours of ICU admission were also collected. Additionally, duration of mechanical ventilation, ICU length of stay, hospital length of stay, opioid use, sedatives use, mortality, use of continuous renal replacement therapy, use of intra-aortic balloon pumping, use of extra corporeal oxygen exchange during ICU stay were recorded. Sepsis was defined as the presence of an infectious disease diagnosis and an elevated SOFA score of two or more points [15]. Benzodiazepines use was defined as continuous intravenous benzodiazepines for more than 24 hours. Mental disorders comprised any mental illness, including schizophrenia and depression. However, dementia was not considered as a mental disorder in this study because we used it as a covariate in the analysis. 

P17, Line 256

bMental disorder; schizophrenia, depression, bipolar disorder, alcoholism, adjustment disorder, and panic disorder.

Comment

6.How did the authors address the patients with dementia or psychiatric disorders in the ICDSC scale? I wonder if those patients should be excluded from the analysis.

Reply

Thank you for pointing this out.

As you mentioned, a past study using ICDSC indicated that false positives of delirium included dementia and mental disorders (Bergeron et al., 2001). However, a different reported high specificity of ICDSC in studies that investigated the detection of delirium (specificity:95%,95%CI:87~98%) (van Eijk et al., 2009). Therefore, we did not think we needed to exclude patients with dementia and mental disorders. Additionally, if similar specificity of ICDSC between before and after phase, the result would have unchanged. 

However, as you pointed out, false positives of delirium may have occurred. Therefore, we added a sensitivity analysis excluding patients with dementia and mental disorders from the primary analysis to the supplementary. The results showed no association between the no-visitation policy and the number of days until the development of delirium (Hazard ratio 0.890, 95%Cl 0.572-1.385). Thank you to your suggestion, we have conducted additional analysis and gained new insights. We also show the revised manuscript below with the modified parts in red in the sensitivity analysis section.

References

Bergeron, N., Dubois, M. J., Dumont, M., Dial, S., & Skrobik, Y. (2001). Intensive care delirium screening checklist: Evaluation of a new screening tool. Intensive Care Medicine, 27(5), 859–864.doi:

10.1007/s001340100909

van Eijk, M. M. J., van Marum, R. J., Klijn, I. A. M., de Wit, N., Kesecioglu, J., & Slooter, A. J. C. (2009). Comparison of delirium assessment tools in a mixed intensive care unit. Critical Care Medicine, 37(6), 1881–1885. doi: 10.1097/CCM.0b013e3181a00118

------Revised Manuscript------

P13, Line 221

Third, a previous study reported the occurrence of false positives when the ICDSC was used to determine delirium in patients with dementia and mental disorders [16]. Therefore, we conducted an analysis using Cox proportional- hazards model, excluding patients with dementia or mental disorders. All covariates were the same as the ones in the primary analysis except for dementia. 

Comment

7. I wonder what is the real impact of the higher percentage of opioids use and the higher number of mechanical ventilation days in the no visitation group. As the authors state it as a study limitation, could it be possible to perform a subanalysis excluding such a difference?

Reply

Thank you for pointing this out.

 As you pointed out, the no-visitation group tended to have higher rates of opioid use and longer duration of mechanical ventilation. Our analysis has not been able to adjust for this difference. For this reason, we changed the variable used in the sensitivity analysis from the mechanical ventilation use to duration of mechanical ventilation. In addition, 84.6% of the patients on ventilators were on opioids. Contrastingly, 86.8% of the patients on opioids were on mechanical ventilators. Thus, we avoided including the opioid variable in our sensitivity analysis due to the possibility of multicollinearity.

　When we performed the Cox proportional-hazards model by changing the covariate of mechanical ventilation use in the primary analysis to duration of mechanical ventilation, the results did not change significantly, with an AHR of 0.938 (95% CI: 0.644-1.367, p-value: 0.741) for the no-visitation policy. In other variables, dementia, APACHE II-age increased the risk of developing delirium. In addition, we agree with your comment and have performed a Cox proportional- hazards model by changing the covariate of mechanical ventilation use in the primary analysis to opioid use. The results did not change significantly, with an AHR of 0.883(95% CI: 0.604-1.291, p-value: 0.520) for the no-visitation policy. The results are shown below.

 Thank you again for pointing this out; we have gained a new insight and confirmed the robustness of the results. We considered duration of mechanical ventilation to be an important variable as well. Therefore, we modified the section on sensitivity analysis and added the table of sensitivity analysis to the supplementary files. Below, we show the revised parts from the manuscript in red.

Estimates of the hazard ratios of variables on development of delirium in the Cox proportional hazards models include covariate as Opioid use 

Variable Adjusted

hazard ratio 95% CI p-value

No-visitation policy 0.883 0.604-1.291 0.520

Age 0.998 0.985-1.013 0.876

Male 1.153 0.780-1.705 0.473

Dementia 1.941 1.164-3.238 0.010

Emergency surgery 1.346 0.717-2.529 0.354

APACHE IIa 1.036 1.011-1.063 0.004

Benzodiazepine

use 1.041 0.572-1.895 0.893

Opioid use 1.789 1.086-2.949 0.022

aAPACHEⅡ score was calculated without age related score

------Revised Manuscript------

P13, Line 221

Second, in order to remove the effects of duration of mechanical ventilation on our findings, we changed the covariate of mechanical ventilation use as binomial variable to the duration of mechanical ventilation, and conducted the Cox proportional-hazards model. All covariates included in the primary analysis were also included int the model, with the exception of mechanical ventilation use. 

Comment

8. In line 252-253, the authors talk about the median number of days for onset of delirium.Among these days, do the authors take into account also the days the patients were sedated?

Reply

Thank you for pointing this out.

In the sections you mentioned, the median and interquartile range of the number of days until the onset of delirium among only patients who developed delirium were listed regardless of whether the patient had been sedated or not. As suggested, these points were difficult to understand. Below is the revised manuscript with the changed parts in red.

------Revised Manuscript------

P18, Line 270

The median number of days for onset of delirium among patients who developed delirium before and after the implementation of the no-visitation policy was not significantly different (2 [IQR], 2–3 vs 2 [IQR], 2–3, p=0.696). This result does not take into account whether patients had been sedated or not. The Kaplan-Meier curve between the before and after phases for the development of delirium is shown in Fig 2. The log-rank test did not show a significant difference between the two phases (p=0.61).

---

## [Decision Letter · Decision Letter 1]

23 Feb 2022

Relationship between no-visitation policy and the development of delirium in patients admitted to the intensive care unit

PONE-D-21-25576R1

Dear Dr. Unoki,

We’re pleased to inform you that your manuscript has been judged scientifically suitable for publication and will be formally accepted for publication once it meets all outstanding technical requirements.

Kind regards,

Andrea Ballotta

Academic Editor

PLOS ONE

Additional Editor Comments (optional):

On the basis of the reviewer's comments the paper can be accepted for publication. Congratulations

Reviewers' comments:

Reviewer's Responses to Questions

**Comments to the Author**

1. If the authors have adequately addressed your comments raised in a previous round of review and you feel that this manuscript is now acceptable for publication, you may indicate that here to bypass the “Comments to the Author” section, enter your conflict of interest statement in the “Confidential to Editor” section, and submit your "Accept" recommendation.

Reviewer #1: All comments have been addressed

Reviewer #2: All comments have been addressed

2. Is the manuscript technically sound, and do the data support the conclusions?

Reviewer #1: Yes

Reviewer #2: Partly

3. Has the statistical analysis been performed appropriately and rigorously? 

Reviewer #1: Yes

Reviewer #2: Yes

4. Have the authors made all data underlying the findings in their manuscript fully available?

Reviewer #1: Yes

Reviewer #2: Yes

5. Is the manuscript presented in an intelligible fashion and written in standard English?

Reviewer #1: Yes

Reviewer #2: Yes

6. Review Comments to the Author

Reviewer #1: The authors have properly addressed my issues. Moreover, a detailed description of what the authors did has been provided. I have no more comments.

Reviewer #2: The Author extensively addressed my comments.

I would underline in the "Limitations of the study" section the fact that the patients of the after phase group had higher mechanical ventilation use rates and longer duration of mechanical ventilation (line 305) hence greater use of sedatives.

7. PLOS authors have the option to publish the peer review history of their article (what does this mean?). If published, this will include your full peer review and any attached files.

Reviewer #1: **Yes: **Mario Mazza

Reviewer #2: No

---

## [Editor Report · Acceptance letter]

28 Feb 2022

PONE-D-21-25576R1 

Relationship between no-visitation policy and the development of delirium in patients admitted to the intensive care unit 

Dear Dr. Unoki:

I'm pleased to inform you that your manuscript has been deemed suitable for publication in PLOS ONE. Congratulations! Your manuscript is now with our production department. 

Kind regards, 

on behalf of

Dr. Andrea Ballotta 

Academic Editor

PLOS ONE